# Response to Ovarian Stimulation for Urgent Fertility Preservation before Gonadotoxic Treatment in *BRCA*-Pathogenic-Variant-Positive Breast Cancer Patients

**DOI:** 10.3390/cancers15030895

**Published:** 2023-01-31

**Authors:** Lina El Moujahed, Robin Philis, Michael Grynberg, Lucie Laot, Pauline Mur, Noemi Amsellem, Anne Mayeur, Alexandra Benoit, Sophia Rakrouki, Christophe Sifer, Maeliss Peigné, Charlotte Sonigo

**Affiliations:** 1Department of Reproductive Medicine and Fertility Preservation, Université Paris-Saclay, Assistance Publique-Hôpitaux de Paris, Antoine Beclere Hospital, 92140 Clamart, France; 2Department of Reproductive Medicine and Fertility Preservation, Université Sorbonne Paris Nord, Assistance Publique-Hôpitaux de Paris, Jean Verdier Hospital, 93143 Bondy, France; 3Service de Biologie de la Reproduction—CECOS, Université Paris-Saclay, Assistance Publique-Hôpitaux de Paris, Antoine Beclere Hospital, 92140 Clamart, France; 4Department of Biology of Reproduction and CECOS, Université Sorbonne Paris Nord, Assistance Publique-Hôpitaux de Paris, Jean Verdier Hospital, 93143 Bondy, France; 5Inserm, Physiologie et Physiopathologie Endocrinienne, Université Paris-Saclay, 94276 Le Kremlin-Bicêtre, France; 6Department of Reproductive Medicine, Hôpital Antoine Béclère, 157 Avenue de la Porte Trivaux, 92140 Clamart, France

**Keywords:** *BRCA* pathogenic variant, controlled ovarian hyperstimulation, fertility preservation

## Abstract

**Simple Summary:**

*BRCA 1/2* pathogenic variants increase the risk of developing early and aggressive breast cancers. For these patients, fertility potential can be directly affected by oncologic treatments. In order to improve their chances of conception after the completion of cancer treatments, fertility preservation should be proposed before the administration of gonadotoxic drugs. The present investigation aims to assess ovarian response to ovarian hyperstimulation in *BRCA 1/2* pathogenic variant carriers. A total of 311 breast cancer patients with known *BRCA* status were included in this retrospective cohort study. The oocyte maturation rate and the number of mature oocytes obtained were significantly lower in the *BRCA*-mutated patients.

**Abstract:**

*BRCA 1/2* pathogenic variants increase the risk of developing early and aggressive breast cancers (BC). For these patients, fertility potential can be directly affected by oncologic treatments. In addition, evidence indicates that *BRCA*-mutated women had a significant reduction in their ovarian reserve. In order to improve their chances of conception after the completion of cancer treatments, fertility preservation should be proposed before the administration of gonadotoxic drugs, ideally by oocyte vitrification after controlled ovarian hyperstimulation (COH). The present investigation aims to assess the ovarian response to COH in *BRCA 1/2*-pathogenic-variant carriers diagnosed with BC. Patient characteristics and COH outcomes were compared between *BRCA*-positive (*n* = 54) and *BRCA*-negative (*n* = 254) patients. The number of oocytes recovered did not differ between the two groups. However, the oocyte maturation rate and the number of mature oocytes obtained (7 (4.5–11.5) vs. 9 (5–14) oocytes, *p* = 0.05) were significantly lower in the *BRCA*-mutated patients. Although individualized COH protocols should be discussed, *BRCA*-mutated patients would benefit from FP before BC occurs, in order to cope with the potential accelerated decline of their ovarian reserve, optimize the success rate of FP by repeating COH cycles, and to preserve the feasibility of PGT-M by collecting a large amount of eggs.

## 1. Introduction

Pathogenic variants in the *BRCA 1* and *BRCA 2* genes, belonging to DNA double-strand-break repair genes, place female carriers at risk of developing breast or ovarian cancer. These cancers are usually more aggressive than in non-mutated females and often occur prior to menopause, sometimes before parental project completion [1,2,3]. Indeed, 12% of the incidences of breast cancer (BC) before 40 years of age occur in *BRCA*-pathogenic-variant carriers. Personalized screenings, early cancer diagnoses, targeted therapies, individualized cancer treatments, and prophylactic strategies have dramatically improved the life expectancy of this patient group, even in cases of previous BC. However, BC treatments often impair the fertility potential of young women and few BC survivors conceive following the completion of anticancer treatment [4,5,6,7,8,9,10]. Oncofertility counseling and further fertility preservation (FP) techniques should be included in the management of BC in young women, whatever their *BRCA* status. Although several FP techniques are currently available, controlled ovarian hyperstimulation (COH) followed by the cryopreservation of mature oocytes is the gold standard and should be performed when possible [11].

Another reproductive issue for *BRCA*-pathogenic-variant carriers is the risk of pathogenic variant transmission to offspring due to its autosomal dominant mode. Preimplantation genetic testing for monogenic disease (PGT-M) may be proposed to avoid the transmission of this critical, hereditary disease by selecting embryos for uterine transfer. The number of mature oocytes obtained is the most important predictive factor for live birth after PGT-M.

Nevertheless, several lines of evidence indicate that the inability of mutated *BRCA* genes to repair DNA double-strand breaks leads to impaired ovarian function due to oocyte aging, apoptosis, and meiotic errors [12,13,14]. In accordance, several studies showed that, even in the absence of BC, female *BRCA*-pathogenic-variant carriers display a low ovarian reserve and a shortened length of reproductive life [15,16,17,18,19]. Several studies have tried to determine if *BRCA* pathogenic variants could impact the *BRCA*-mutated patients’ response to COH for FP and/or PGT-M contexts. Results were contradictory [16,20,21,22,23,24,25,26], based on small populations, and were not always adjusted to the presence or lack thereof of BC.

The aim of the present study is to assess the ovarian response to COH in *BRCA 1/2*-pathogenic-variant carriers diagnosed with BC who are undergoing oocyte cryopreservation before gonadotoxic oncological BC treatment.

## 2. Material and Methods

### 2.1. Patients

Between July 2013 and July 2021, all BC females (*n* = 464) aged between 18 and 43 years who underwent COH for FP in the Jean Verdier or Antoine Béclère reproductive medicine and FP centers were screened for eligibility for this retrospective cohort study. Inclusion criteria were a histologically confirmed diagnosis of BC, known *BRCA* status, and having undergone COH in the context of emergency FP with oncologists’ authorization. Exclusion criteria were the absence of a *BRCA* pathogenic variant test or results and the absence of oocyte pick-up or other FP technique (oocyte cryopreservation after in vitro maturation or ovarian cortex cryopreservation). A total of 311 females were included. Among them, 57 had a known *BRCA* pathogenic variant and 254 did not. All patients provided their informed consent to perform COH and oocyte and/or embryo vitrification.

### 2.2. Hormonal Measurements and Ultrasound Scans

All participants underwent a random assessment of their ovarian reserve by measurement of their serum-anti-Müllerian-hormone (AMH) levels and an ultrasound scan for antral follicle counting at the time of oncofertility counseling. Serum-AMH levels were determined using an ultrasensitive, enzyme-linked immuno-sorbent assay (ELISA). Ultrasound scans were performed using a 3.7–9.3 MHz. multi-frequency transvaginal probe (RIC5-9H, Voluson E8, General Electric Medical Systems, Paris, France) to evaluate the number and sizes of small antral follicles. All follicles measuring 2–9 mm in mean diameter (mean of two orthogonal diameters) in each ovary were considered for AFC.

### 2.3. Controlled Ovarian Hyperstimulation Protocol, Oocyte Retrieval Technique, and Vitrification Procedure

All patients underwent a GnRH antagonist random start protocol for ovarian stimulation [27]. The exogenous FSH starting dose was individualized (between 150 and 450 IU/d) according to patients’ clinical characteristics and ovarian reserve tests. Some patients received letrozole supplementation during FSH administration. The GnRH antagonist administration was initiated between the sixth and eighth days of exogenous gonadotropin stimulation. Daily FSH doses were adjusted according to E_2_ levels and/or the number of growing follicles. During the last days of COH, patients had daily visits at our Institution for ultrasonographic and hormonal examinations to define the proper timing for ovulation trigger (OT), when at least three preovulatory follicles (16–22 mm in diameter) were observed. The administration of hCG (250 μg of Ovitrelle, Merck Serono, Lyon, France) or GnRH agonist (triptorelin 0.1 mg, Decapeptyl^®^, Ipsen Pharmaceuticals, 0.2 mg, S.C.) was decided according to the estimated risk of ovarian hyperstimulation syndrome. Overall, when less than 20 follicles > 11 mm were observed, the trigger was performed using hCG. The GnRH agonist was administered when more than 20 follicles > 11 mm were present within the ovaries. In case of GnRH-agonist triggering, the LH surge was checked by the measurement of blood-LH levels approximately twelve hours after the injection. Oocytes were retrieved 36–38 h after the ovulation trigger under general or local anesthesia using a single-lumen. 19-gauge needle (K-OPS-7035-Wood; Cook, France) guided by vaginal ultrasound. No follicular flushing was performed.

Follicular fluid with cumulo-ovocyte complexes (COCs) was collected in 15 mL NucleonTM tubes (Nunc A/S, Roskilde, Denmark) and then dispersed into NucleonTM culture dishes (Nunc A/S). After washing in Universal IVF Medium^®^ (Origio, Versailles, France), COCs were denuded with a hyaluronidase solution (Syn Vitro Hyadase, Origio, Versailles, France). While immature eggs were discarded, mature oocytes (having reached metaphase II (MII) stage) were vitrified or inseminated by ICSI before vitrification at the cleavage or blastocyst stage.

### 2.4. Evaluated Parameters

Patients, tumor and COH characteristics, and COH outcomes were collected retrospectively from medical files. Oocyte retrievals and maturation rates were calculated from the ratios of the number of retrieved oocytes × 100/AFC and the number of MII oocytes × 100/number of retrieved oocytes, respectively. In addition, follicular responsiveness to FSH was assessed by the follicle output rate (FORT) index, calculated by the number of follicles 16–22 mm in diameter on the day of the ovulation trigger × 100/AFC [28].

### 2.5. Statistical Analysis

The results are presented as median and interquartile ranges (25–75%). Comparisons between groups (*BRCA* and non-*BRCA*) were performed using Mann–Whitney tests for continuous variables and chi-square tests (or the Fisher exact test when appropriate) for categorical variables. When patients underwent more than one COH cycle, we analyzed the outcomes of the first cycle only. A *p* value < 0.05 was considered statistically significant. All statistics were performed using NCSS Statistical Software (2021) (NCSS 2021 Statistical Software (2021). NCSS, LLC. Kaysville, UT, USA, ncss.com/software/, accessed on 27 November 2022).

### 2.6. Ethical Approval

Consent for the use of their medical data for research purpose was obtained from all patients/couples at the time of the initiation of the FP procedure. Our database was approved by the National Data Protection Authority (Commission Nationale de l’Informatique et des Libertés, CNIL no. 1217921). An Institutional Review Board approval was obtained for this retrospective study (CLEA 2022-264).

## 3. Results

### 3.1. Females and Cancer Characteristics

A total of 311 BC females having undergone COH for FP before adjuvant or neoadjuvant chemotherapy were included in the present study. Among them, 57 carried a *BRCA* pathogenic variant (*n* = 36 and *n* = 21 for *BRCA 1* and *BRCA 2*, respectively). The diagnosis of a *BRCA* pathogenic variant was mainly made after the completion of the FP procedures. The mean age of the entire cohort at the beginning of the first COH for FP was 33.3 ± 3.9 years.

Overall, *BRCA*-mutated and non-mutated patients were comparable in terms of age (33.4 (30.5–36.0) vs. 33.2 (30.5–36.2) years, respectively, *p* = 0.77). Although AFC was not significantly different between both groups (16 (12–25) vs. 19 (2–27), *p* = 0.35), serum-AMH levels were significantly lower for the *BRCA*-mutated females (1.6 (0.8–2.9) vs. 2.4 (1.4–3.8) ng/mL, *p* = 0.02) (Table 1).

The majority of BC patients (95.4%) had ductal carcinoma. As expected, *BRCA*-mutations were associated with more aggressive cancer parameters, including a higher percentage of grade III tumors (66.7% vs. 50.7%, *p* = 0.04) and a higher proliferation index Ki 67 (50% (22.5–70.0) vs. 30% (15–58.8), *p* = 0.004, respectively) when compared to non-mutated patients. Moreover, the prevalence of triple-negative tumors was significantly higher in mutated females (64.3% vs. 18.1%, *p* < 0.0001) (Table 1).

### 3.2. Controlled Ovarian Hyperstimulation Outcomes According to BRCA Status

Only the first COH cycle was included in the analysis (17 women underwent more than one COH cycle). Overall, the proportion of females having begun ovarian stimulation during the follicular or luteal phase was comparable in both groups (56.9% vs. 58.1% in the follicular phase, *p* = 0.87), with similar FSH starting doses, COH duration, and total amount of exogenous gonadotropins. The mean number of oocytes recovered as well as the FORT index did not significantly differ between the *BRCA*-mutated and non-mutated groups (10 (6–15) vs. 11 (7–17) oocytes, *p* = 0.16; FORT 33.3% (23.1–48.3) vs. 29.4% (17.9–47.3), *p* = 0.27, respectively). Oocyte maturation rates were significantly altered in the *BRCA*-mutated group in comparison to the non-mutated group (78.6% (33.1–96.4) vs. 85.7% (70.0–100), respectively, *p* = 0.04), leading to a lower number of MII oocytes (7 (4.5–11.5) vs. 9 (5.0–14.0) MII respectively, *p* = 0.049). In both groups, the large majority of females chose the vitrification of oocytes instead of embryos (Table 2).

### 3.3. Controlled Ovarian Hyperstimulation Outcomes According to the Type of BRCA Pathogenic Variant

Table 3 describes cancer characteristics and COH outcomes according to the type of variant (*BRCA 1* or *BRCA 2*). Age, AFC, and AMH-serum level were not significantly different between both groups. The proportion of triple-negative tumors was significantly higher for *BRCA 1*-mutated women than *BRCA 2*-mutated females (86% vs. 25%, *p* < 0.0001), as well as the Ki 67 proliferation index (*p* = 0.002). However, no parameter of COH was significantly different according to the type of *BRCA* pathogenic variant (Table 3).

## 4. Discussion

The present investigation aimed to assess whether *BRCA 1/2* pathogenic variants affect the response to COH for urgent FP before BC gonadotoxic treatment. We failed to find any difference for the total dose of gonadotropins, number of retrieved oocytes, oocyte retrieval rates, or FORT index. However, a significantly lower maturation rate and a significantly lower number of mature oocytes were observed in *BRCA*-pathogenic-variant carriers. The comparison between *BRCA 1* and 2 groups showed no significant difference in COH response and outcomes.

Over the past years, several studies have analyzed the characteristics of ovarian stimulation in *BRCA*-mutation carriers with conflicting results [16,20,21,22,23,24,25,26]. As the *BRCA* pathogenic variant is a relatively rare condition, all these studies are characterized by the relatively low number of patients, ranging from 10 to 81. Moreover, COH indications (FP for BC or COH for PGT-M) as well as the population used for control group (FP for BC, IVF with PGT-M for other reasons) imply the interpretations of results with caution.

Our study did not show any difference in the number of oocytes retrieved between *BRCA*-mutated and non-carriers, which is in accordance with some investigations [20,22,24,25], while others publications reported a decreased number oocytes obtained in mutated patients [16,21,23,26]. A recent a meta-analysis suggested that the *BRCA* pathogenic variant did not significantly affect the number of oocytes yielded after COH (*p* = 0.77), despite a high and statistically significant heterogeneity with an I2 value of 75% (*p* = 0.02) [29]. Interestingly, our findings indicated that the *BRCA* mutation was associated with a significantly reduced oocyte maturation rate and a lower number of mature oocytes. Similar results were reported by Porcu et al. in a study including 22 *BRCA* pathogenic variants carriers compared to 24 BC non-mutated patients [25]. However, two others studies reporting a similar number of retrieved oocytes failed to evidence significant differences in the number of matured oocytes [22,24]. Nevertheless, in the study by Lambertini et al., although the indication for COH was also FP for BC in both groups the number of patients included was limited (less than 20 in each group) [22]. In the other study, *BRCA*-mutated patients with or without BC were compared with a highly heterogenous control group including women with different types of malignancies or without pathology, which might explain the results [24].

The reduction in oocyte maturation during COH might find its explanation directly from the recognized role of *BRCA* genes in DNA repair and homologous combination [30]. As these processes are required during the meiotic process, especially during prophase I, it has been suggested that the *BRCA* pathogenic variant induces an alteration of ovarian reserve through the anticipated apoptosis of oocytes [16]. Several lines of evidence indicate that *BRCA*-mutated patients had lower levels of AMH when compared to non-*BRCA* populations [17,18,24,31,32,33], and a recent meta-analysis confirmed this data [34] with an AMH that was 1.0 ng/mL lower in *BRCA*-mutated patients before FP for BC compared to non-carriers with BC. Our results are in accordance with these results, as serum-AMH levels were significantly lower in the *BRCA*-pathogenic-variant carriers. However, more robust data is still needed to clarify the actual impact of *BRCA* mutation on ovarian function, follicular status, and COH outcomes.

It has recently been hypothesized that some BC prognostic factors (such as a triple-negative tumor, high Ki67 proliferation index, or SBR grade) could be an independent cause of poor response to COH. Therefore, our results might be explained by the tumoral aggressivity of cancers observed in case of *BRCA* mutation [35]. Indeed, it is well established that this subgroup of patients present with more aggressive tumors [36,37,38,39,40], which was also observed in our study. This may be presumably explained by the function of *BRCA* genes in the process of oncogenesis via their roles in DNA repair and homologous recombinations [30]. Thus, to clarify if our findings are linked to *BRCA* or BC status, it would be interesting to compare ovarian reserve and COH outcomes between *BRCA*-pathogenic-variant carriers with and without BC, and non-*BRCA*-mutated patients with and without BC.

Whatever the results, FP should be considered in the case of *BRCA* mutation, even before BC occurrence. Indeed, non-urgent FP in this context may avoid the possible impact of breast malignancy on COH outcomes, allow for multiple stimulation cycles to increase the number of frozen eggs, and further the overall results of FP and PGT-M, if any.

We also compared BC characteristics and ovarian response to gonadotropins according to the type of *BRCA* mutation (*BRCA 1* or *2*). In 2008, Atchley et al. reported a difference in BC severity between these two groups [36], and our study provides additional insight in this regard. As a matter of fact, the *BRCA 1* group presented more frequently with triple-negative tumors and higher histopronostic grades. We failed to observe any difference in ovarian reserve markers or response to COH between the two subgroups, which is concordant with the study by Gunnala et al, who found a comparable number of oocytes retrieved (15.5 ± 8.1) vs. 17.1 ± 5.9) oocytes, *p* = 0.5) and AMH levels ((2.4 ± 1.7) vs. 3.6 ± 2.4) ng/mL, *p* = 0.06) in *BRCA 1*- and *BRCA 2*-mutated women, respectively [24]. Porcu et al. [25] found lower serum-AMH levels and numbers of mature oocytes in *BRCA 1*-mutated patients in a very small sample.

We report the largest study assessing ovarian response to COH in *BRCA*-pathogenic-variant BC females. Another strength of our investigation is the homogeneity of the population, including only women with a known *BRCA* status. The retrospective design and the lack of data concerning fertilization rates and oocyte quality are the main limitations of the present study. As a matter of fact, these latter variables determine the ability to obtain embryos suitable for transfer and, ultimately, live births.

## 5. Conclusions

In conclusion, in this large study assessing COH in a context of urgent FP before BC gonadotoxic treatment, a *BRCA* pathogenic variant does not affect the response to COH in terms of number of oocytes retrieved, FORT, or oocyte retrieval rates but may alter the capacity of oocytes to reach the MII stage. Unfortunately, this scientific contribution is difficult to apply in clinical practice, as *BRCA* status is often unknown at the time of urgent FP. Nevertheless, when *BRCA* status is known before COH, individualized ovulation trigger might be considered to improve the oocyte maturation rate. Moreover, our study emphasizes that *BRCA*-mutated women should receive systematic FP counseling for considering oocyte vitrification to cope with the potential accelerated decline of their ovarian reserve, to optimize the success rate of FP by repeating COH cycles, and to preserve the feasibility of PGT-M by collecting a large amount of eggs.

## Figures and Tables

**Table 1 cancers-15-00895-t001:** Females’ characteristics.

	*BRCA* (*n* = 57)	Non-*BRCA* (*n* = 254)	*p*
Females’ characteristics			
Age (years) ^a^	33.4 (30.5–36)	33.2 (30.5–36.2)	0.77
AFC (follicles)	16 (12–25)	19 (12–27)	0.35
AMH (ng/mL) ^a^	1.6 (0.8–2.9)	2.4 (1.4–3.7)	0.02
Breast cancer characteristics			
Ductal BC	54/57 (94.7%)	240/251 (95.6%)	1
Triple-negative	36/56 (64.3%)	44/243 (18.1%)	0.0001
Ki 67 (%)	50 (22.5–70)	30 (15–58.8)	0.004
Grade III	34/51 (66.7%)	111/219 (50.7%)	0.04

^a^ Median (25–75%); SD: standard deviation; BC: breast cancer.

**Table 2 cancers-15-00895-t002:** COH characteristics and outcomes.

	*BRCA* (*n* = 57)	Non-*BRCA* (*n* = 254)	*p*
Start of stimulation			0.14
Luteal phase	22/51 (43.14%)	88/210 (41.9%)	
Follicular phase	29/51 (56.9%)	122/210 (58.1%)	
Protocol			0.40
Conventional antagonist	41 (71.9%)	196 (77.2%)	
Antagonist + letrozole	16 (28.1%)	58 (22.8%)	
Starting dose of gonadodropins (IU) ^a^	300 (225–350)	300 (225–350)	0.37
Duration of stimulation (days) ^a^	10 (9–12)	10 (9–12)	0.57
Total dose of gonadotropins (IU) ^a^	2700 (2400–3937)	3000 (2062–3600)	0.73
Serum E2 levels on dOT ^a^	587 (310–1227)	729 (334–1487)	0.15
Triggering type			0.14
Recombinant hCG	20/52 (38.5%)	58/208 (27.9%)	
GnRH agonist	32/52 (61.5%)	150/208 (71.1%)	
No of follicles ≥ 16 mm on dOT ^a^	6 (4–7)	5 (4–8)	0.75
No of retrieved oocytes ^a^	10 (6–15)	11 (7–17)	0.16
No of metaphase II oocytes ^a^	7 (4.5–11.5)	9 (5–14)	0.05
Oocyte retrieval rate (%)	63.6 (40.0–80.1)	58.8 (38.5–83.3)	0.95
Oocyte maturation rate (%)	78.6 (53.1–96.4)	85.7 (70–100)	0.04
FORT * (%)	33.3 (23.1–48.3)	29.4 (18–48)	0.27

^a^ Median (25–75%); * FORT (follicular output rate) = (No of follicles > 16 mm on dOT/No of antral follicles on d0) × 100; Maturity rate: =(N° of MII oocytes/N° of retrieved oocytes) × 100; Oocyte retrieval rate = (N° of retrieved oocytes/No of antral follicles on d0) × 100; dOT: day of triggering, d0: day of starting.

**Table 3 cancers-15-00895-t003:** Females, breast cancer, COH characteristics and outcomes according to the type of *BRCA* pathogenic variant.

	*BRCA 1* Mutation (*n* = 36)	*BRCA 2* Mutation (*n* = 21)	*p*
Females’ characteristics			
Age (years) ^a^	34.1 (35.4–36.2)	32.3 (29.7–35.0)	0.25
AFC (follicles)	15 (11.0–21.5)	20 (14–31)	0.21
AMH (ng/mL) ^a^	1.6 (0.8–2.9)	2.4 (1.4–3.7)	0.27
Breast cancer characteristics			
Ductal BC	34/36 (94.4%)	20/21(95.2%)	1
Triple-negative	31/36 (86.1%)	5/20 (25%)	<0.0001
Ki 67 (%)	60 (40–80)	20 (14–50)	0.002
Grade III	25/31 (80.7%)	10/20 (45%)	0.01
*COH characteristics*			
Start of stimulation			
Luteal phase	14/32 (43.7%)	8/19 (42.1%)	0.9
Follicular phase	18/32 (56.3%)	11/19 (57.9%)	
Protocol			
Conventional antagonist	28/36 (77.8%)	13/21 (61.9%)	0.19
Antagonist + letrozole	8/36 (22.2%)	8/21 (38.1%)	
Starting dose of gonadotropins (IU) ^a^	300 (225–350)	300 (237–325)	0.43
Duration of stimulation (days) ^a^	10.9 ± 2.7 10 (9–12)	9.9 ± 1.2 10 (9–12)	0.24
Total dose of gonadotropins (IU) ^a^	2962.5 (2400–3994)	2625 (2213–3750)	0.35
Serum E2 levels on dOT (pg/mL) ^a^	637.7 (340–1336)	480 (240–932)	0.24
Triggering type			0.68
Recombinant hCG	13/32 (40.6%)	7/20 (35%)	
GnRH agonist	19/32 (59.4%)	13/20 (65%)	
No of follicles ≥ 16 mm on dOT ^a^	6 (4–7)	6.5 (4–8)	0.08
No of oocytes retrieved (n) ^a^	9 (5–15)	14 (7–16)	0.09
No of metaphase II oocytes (n) ^a^	7 (3–11)	9 (5–13)	0.08
Oocyte Retrieval Rate (%)	55.0 (32.8–76.9)	68.8 (43.8–92.7)	0.14
Oocyte maturation rate (%)	78.2 (50.1–97.9)	80.0 (69.1–96.4)	0.39
FORT * (%) ^a^	33.3 (21.3–48.1)	32.5 (24.8–50.5)	0.85

^a^ Median (25–75%); * FORT (follicular output rate) = (No of follicles > 16 mm on dOT/No of antral follicles on d0) × 100; Maturity rate: =(N° of MII oocytes/N° of retrieved oocytes) × 100; Oocyte retrieval rate = (N° of retrieved oocytes/No of antral follicles on d0) × 100; dOT: day of triggering, d0: day of starting.

## Data Availability

The data available in this review article are available in the present manuscript.

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
