# Peer review of "Response to Ovarian Stimulation for Urgent Fertility Preservation before Gonadotoxic Treatment in BRCA-Pathogenic-Variant-Positive Breast Cancer Patients"

_cancers, 2023, doi:10.3390/cancers15030895_

Round 1

Reviewer 1 Report

This paper evaluates the different outcomes in terms of oocytes response to ovarian stimulation in BC patients BRCA +ve. Overall the topic is very interesting altough the results are not surprising since other studies have already shown the reduced in overall ovarian performance.

This study, however focuses on the reduction in oocytes maturation which has not been shown before. At this regard I have some concerns as the design of the study (retrospective not prospective and not randomised) and the heterogeinicity in the different protocols to stimulate the ovaries and to trigger the ovulation might have played a role in affecting the final outcome (oocytes' maturation). For instance the authors said in line 103 that the dose of stimulation was individualised ed between 150-450IU but in table 2, it says starting dose was 300IU in both groups. Could this be clarified? Why some patients were given agonist trigger and some Ovitrelle? Can the trigger criteria and choice of trigger be explained in  more details? Why some people were given letrozole? Again, can this be explained in the material and methods section? more info on oocytes pick ups are needed? flush or no flush? single or double lumen? rate of OHSS? 

More remarks below:

line 255 remove of before present

line 289: please explain the meaning of individualised protocols

Thank you 

Author Response

We thank the two reviewers for the diligent and insightful review of our manuscript. We have responded to each comment in the cover letter and have adapted the manuscript accordingly. The changes are highlighted in the revised manuscript with the 'track changes' tool in MS Word.

Thank you very much for your kind consideration of our work.

We hope the reviewers will now find our manuscript acceptable for publication.

Yours truly,

Charlotte Sonigo

Reviewer 2 Report

In this study, the authors assessed ovarian response to ovarian hyperstimulation in BRCA1/2 pathogenic variants carriers and the results showed that although there is no significant differences of oocytes retrieved between BRAC carries and the non-BRAC female, the oocyte maturation rate and the number of mature oocytes obtained were significantly lower in the BRAC-mutated patients. Here are some concerns need to be clarified.

1. Because the slight difference of ovarian hyperstimulation protocol for each individual will cause the different outcome of ovarian response, how did the author suppress these interference of the protocol in this study?

2. What was the criteria for selecting the BRCA pathogenic variants in this study? How could the author know the breast cancer was caused by the BRCA variants instead of other potential genetics pathogenic variants? More details of the genetic pathogenic variants selecting criteria need to be addressed.

3. Line 135-136, confused about the analyze for the first cycle only. Could the author give more explanation about this?

4. For discussion part, the embryo/embryonic development quality should also be taken into consideration besides of fertilization rate and oocyte quality.

5. Since the BRCA pathogenic variant does not affect the response to COH in terms of number of oocytes retrieved, more combination protocols of ovarian hyperstimulation are worth of exploration. The author should take this into consideration and discussed more deeply and properly.

Author Response

(The authors gave the same response as above.)

Round 2

Reviewer 2 Report

Overall the points that I raised have been addressed properly. I agree to publish this paper at the present form.